# Role of a ZF-HD Transcription Factor in miR157-Mediated Feed-Forward Regulatory Module That Determines Plant Architecture in *Arabidopsis*

**DOI:** 10.3390/ijms23158665

**Published:** 2022-08-04

**Authors:** Young Koung Lee, Sunita Kumari, Andrew Olson, Felix Hauser, Doreen Ware

**Affiliations:** 1Cold Spring Harbor Laboratory, 1 Bungtown Road, Cold Spring Harbor, NY 11724, USA; 2Institute of Plasma Technology, Korea Institute of Fusion Energy, 37, Dongjangsan-ro, Gunsan-si 54004, Korea; 3Division of Biological Sciences, University of California–San Diego, La Jolla, CA 92093, USA; 4USDA-ARS, Robert W. Holley Center, Ithaca, NY 14853, USA

**Keywords:** ChIP-Seq, feed-forward loop (FFL), gene regulatory network (GRN), HB34, miRNA, next-generation sequencing, shoot branching

## Abstract

In plants, vegetative and reproductive development are associated with agronomically important traits that contribute to grain yield and biomass. Zinc finger homeodomain (ZF-HD) transcription factors (TFs) constitute a relatively small gene family that has been studied in several model plants, including *Arabidopsis thaliana* L. and *Oryza sativa* L. The ZF-HD family members play important roles in plant growth and development, but their contribution to the regulation of plant architecture remains largely unknown due to their functional redundancy. To understand the gene regulatory network controlled by ZF-HD TFs, we analyzed multiple loss-of-function mutants of ZF-HD TFs in *Arabidopsis* that exhibited morphological abnormalities in branching and flowering architecture. We found that ZF-HD TFs, especially HB34, negatively regulate the expression of miR157 and positively regulate SQUAMOSA PROMOTER BINDING–LIKE 10 (*SPL10*), a target of miR157. Genome-wide chromatin immunoprecipitation sequencing (ChIP-Seq) analysis revealed that *miR157D* and *SPL10* are direct targets of HB34, creating a feed-forward loop that constitutes a robust miRNA regulatory module. Network motif analysis contains overrepresented coherent type IV feedforward motifs in the amiR zf-HD and *hbq* mutant background. This finding indicates that miRNA-mediated ZF-HD feedforward modules modify branching and inflorescence architecture in *Arabidopsis*. Taken together, these findings reveal a guiding role of ZF-HD TFs in the regulatory network module and demonstrate its role in plant architecture in *Arabidopsis*.

## 1. Introduction

The architecture of sessile plants is primarily determined by the pattern of shoot branching, which in turn is affected by numerous endogenous, developmental, and environmental factors. Shoot branching is initiated in the axillary meristem (AM) region, a small tissue consisting of undifferentiated cells at the axil of the leaf. The AM promotes the initiation of a few leaves to form an axillary bud, which either turn into a lateral branch or become dormant [1]. The fate of the axillary bud is controlled by many endogenous and developmental signals that are regulated by genetic factors [2,3]. Some developmental transcription factors (TFs) that participate in the differentiation and formation of shoot apical meristem are involved in AM formation and regulate hormonal accumulation [4,5,6]. These TFs, encoded by *SHOOT MERISTEMLESS* (*STM*), *REVOLUTA* (*REV*), and *CUP SHAPED COTYLEDONS* (*CUC*) *1*, *2*, and *3*, are class I KNOTTED-like homeobox TFs containing a StAR-related lipid-transfer lipid-binding domain, an HD-ZIP domain, and a No Apical Meristem NAM domain, respectively [5,6]. Additionally, two MYB transcriptional regulators, LATERAL ORGAN FUSION (LOF) 1 and 2, contribute to AM formation and lateral organ separation and overlap functionally with CUC2, CUC3, and STM [7].

TFs in higher plants, which can be classified into approximately 58 families, play pivotal roles in multiple developmental processes [8]. The family of zinc finger homeodomain TFs (ZF-HD TFs) is smaller than most other TF families. In *Arabidopsis* and other crop plants, these TFs contribute to hormone-related signaling [9,10], responses to environmental stress [11,12], and developmental processes [13,14]. The *Arabidopsis* genome encodes 17 ZF-HD TFs, which can be broadly subdivided into two groups: a larger group consisting of genes encoding ZF-HD TFs with Cys/His-rich dimerization domains at their N-terminal and ZF homeodomains at their C-terminal, and a smaller group of three genes encoding mini-zinc finger (MIF) proteins [15,16]. The MIF proteins contain Cys/His-rich dimerization domains (CX3HX11nCX12–26CX2CXCHX3H) at their N-terminal, but lack the ZF homeodomain and are consequently relatively short [17]. The results of yeast two-hybrid (Y2H) experiments suggest that many ZF-HD TFs form hetero- or homo-dimeric complexes that regulate gene expression and thereby, determine proper organ development [13,18]. However, the transcriptional regulatory network modules controlled by ZF-HD TFs and their direct downstream targets remain to be elucidated.

MicroRNAs (miRNAs) also influence shoot architecture via post-transcriptional repression of their target genes. The NAC (NAM, ATAF1/2, CUC2) domain proteins are plant-specific TFs that contribute to a wide range of developmental processes. A subset of NAC-domain TFs, CUC1, CUC2, NAC1, ORE1, NAC80, and NAC100, are post-transcriptionally regulated by miR164 [19], and as noted above, CUC1 and CUC2 in turn regulate *LAS* [6]. miR164, encoded by *miR164A*, *B*, and *C*, plays specific roles in flower development [20]. *MIR164A/B/C* triple mutants have extra buds with abnormal phyllotaxis in the axil of the cauline leaf. The *CUC1*, *2*, and *3* genes encode NAC domain TFs, and *cuc3-2* mutants have defects in formation of the AM [6]. Thus, CUC3, which is functionally redundant with CUC1 and 2, along with miR164, plays a key role in the initiation of AM formation. miR156 is the master regulator of the vegetative phase change in *Arabidopsis* [21]. The miR 156/157 family, with their highly similar sequences containing only one or two different nucleotides [22], are one of the most conserved miRNA families in the plant kingdom that target *SQUAMOSA PROMOTER BINDING PROTEIN-LIKE* (*SPL*) TFs, which are involved in phase transition, juvenile to adult and flowering transition, control of branching and plant architecture, leaf morphology, fruit ripening and response to abiotic and biotic stress in various plant species [23,24]. In *Arabidopsis*, miR156/157 promotes branching by repressing *SPL9/15* and *SPL2/10/11*, whereas *OsSPL14* in rice, also known *as IDEAL PLANT ARCHITECTURE1* (*IPA1*), and *OsSPL7* were reported as key regulators of tillering and panicle architecture [25]. Overexpression of miR156 negatively regulates SPL genes. The function of the miR156/SPL module in branching is conserved in other crop species [26,27,28]. Many aspects of the biogenesis and targets of miRNAs have been elucidated [29,30,31,32], but the factors that regulate miRNA expression, as well as their biological roles and underlying molecular mechanisms, remain largely undiscovered.

In this study, we identified multiple ZF-HD TFs as novel regulators of plant architecture that have a redundant role for shoot branching and floral formation. We used systems biology approaches to elucidate the gene regulatory network (GRN) that regulates vegetative architecture, as revealed by the enhanced branching phenotype and later effects on flowering patterns in *Arabidopsis*. On the basis of these findings, we propose that HB34, miR157, and SPL10 act together in a feed-forward regulatory module that controls plant architecture in *Arabidopsis*.

## 2. Results

### 2.1. ZF-HD TFs Regulate Plant Architecture in Arabidopsis

MiRNAs play a significant role in post-transcriptional gene regulation through their effects on target mRNA and miRNA genes; thus, their targets and TFs are key regulators. We previously used a gene-centered approach to identify TFs that control the expression of miRNAs involved in development and stress [33]. To understand the topology of GRN, we have classified TF families as in-degree and out-degree with highly connected protein and DNA interactions, called the TF subnetwork. The ZF-HD TF family contains a highly connected subnetwork. To characterize members of the ZF-HD TF family, focusing on the highly connected network of miRNA-mediated regulation of ZF-HD TF family members, we screened T-DNA insertion mutants and obtained T-DNA insertion mutants of HB23, *HB31*, *HB33*, and *HB34* (Figure 1a). Homozygous mutants were isolated by PCR-based genotyping (Appendix A). However, the single loss-of-function mutations caused no observable phenotypic changes during development. To explore the possibility of functional redundancy among ZF-HD TFs, we constructed multiple loss-of-function mutants through genetic crosses of single-mutant lines, generating double (*hb33 hb34*) and quadruple mutants (*hb23 hb31 hb33 hb34*; hereafter, *hbq*). The *hb33 hb34* double mutant had higher vegetative branching number and leaf number, along with reduced leaf size and plant height, than the wild type (WT) (Appendix A). RT-PCR (reverse transcription polymerase chain reaction) analysis revealed that *HB33* and *HB34* were significantly downregulated in the *hb33 hb34* double mutant (Appendix A). At the cellular level, *hb33 hb34* mutant plants had reduced epidermal cell size in their leaves (Appendix A), explaining their smaller leaf size. *hbq* plants had a more pronounced morphological phenotype than the *hb33 hb34* double mutant (Figure 1b–d, Appendix A). In addition, we used an artificial miRNA (amiRNA) approach to generate transgenic lines where multiple ZF-HD TFs (*HB22*, *HB23*, *HB25*, *HB26*, *HB27*, *HB29*, *HB30*, *HB33*, and *HB34*) were targeted [33]. We refer to these two mutants (amiR zf-HD and hbq) as zinc finger homeodomain (*zf-hd*) mutants.

The morphological phenotypes of the amiR zf-HD lines included elevated branching number, reduced silique length, decreased plant height, and distorted floral architecture (e.g., curved and twisted gynoecium and incompletely opened petals) consistent with those of *hbq* (Figure 1b–f and Appendix A). At the cellular level, amiR zf-HD plants had irregular cell shapes in the sepal and reduced cell size in the proximal region of the epidermal petal (Figure 1g–l). In addition, amiR zf-HD and *hbq* exhibited reduced leaf size, plant height, and floral organ difference that impacted silique length (Appendix A). During development, amiR zf-HD and *hbq* plants exhibited delayed initiation of rosette branching on the 25th day after germination, but elevated branching number on the 67th day after germination (Figure 1m). Taken together, these findings show that the ZF-HD TF genes are functionally redundant and play roles in determining vegetative and floral architecture during *Arabidopsis* development.

To elucidate the expression patterns of these ZF-HD genes, we performed qRT-PCR analyses of multiple tissues. The mRNAs encoding *HB31*, *HB33*, and *HB34* were present at low levels in root, leaf, and vegetative tissue, and at much higher levels in floral tissue, especially in young flowers (Appendix A). In RNA-seq for the root and floral tissue, *HB23* had increased expression in the floral tissue compared to that in the root tissue (Appendix A). GENEVESTIGATOR, a publicly available database (http://genevestigator.com/, accessed on 8 June 2020), confirmed that *HB23*, *HB31*, *HB33* and *HB34* have higher expression in the flower including pistil and receptacle tissue than that in the root cell (Appendix A).

RNA-seq revealed that among the 17 ZF-HD TF genes in *Arabidopsis*, 13 (including *HB23*, *HB30*, *HB31*, *HB33*, *HB34*, and *MIF2*) were expressed at higher levels in young flower than in root, whereas the remaining four (*HB24*, *HB29*, *MIF1*, and *MIF3*) were more highly expressed in root than in floral tissue (Appendix A). The RNA-seq data were consistent with expression data obtained from a public database [34] (Appendix A). These patterns suggest that ZF-HD TF genes play their primary functional roles in the development of young flowers. Next, we investigated the subcellular localization of HB23, HB31, HB33, and HB34 using a transient expression assay in which we examined tobacco epidermal cells using fluorescence microscopy. In plants expressing HB23-green fluorescent protein (GFP) or HB34-GFP, we observed strong GFP fluorescence in the nucleus, overlapping the signal from red fluorescent protein (RFP) fused to a nuclear localization sequence (NLS) (Figure 1n,q); this localization pattern is typical of TFs. Intriguingly, both HB31-GFP and HB33-GFP were localized in both the nucleus and cytosol (Figure 1o,p), suggesting that in addition to their transcriptional functions, these TFs play other regulatory roles, perhaps related to environmental stress responses or intracellular signaling pathways [10,35,36]. Based on the gene localization and phylogenetic analysis (Figure 1n–q and Appendix A), ZF-HD genes seem to have distinct functional roles: HB23 and HB34 are involved in transcriptional regulation and HB31 and HB33 appear to be related with abiotic or biotic response or signaling pathway.

### 2.2. ZF-HD TFs Regulate Shoot Architecture in a Dose-Dependent Manner

To elucidate how ZF-HD TFs regulate gene expression, as well as to identify their downstream targets, we performed genome-wide transcriptomic analysis in young floral tissues of amiR zf-HD and *hbq.* Six ZF-HD TF genes, *HB23*, *HB24*, *HB26*, *HB27, HB34,* and *MIF2,* were expressed at lower levels in the amiR zf-HD plants than in WT (Figure 2a and Appendix A).

Transcriptomics analysis revealed that the expression of *HB23*, *HB31*, *HB33,* and *HB34* was dramatically reduced in the *hbq* mutant compared to WT (Figure 2b), and this finding was validated using qRT-PCR (Figure 2c). Put together, these observations confirmed that all of these lines were true loss-of-function mutants in the corresponding members of the ZF-HD TF family.

Hauser et al. (2012) mentioned that overall phenotypes of amiRNA-zfDH have an increased stem number and short silique consistent with the four independent amiR zf-HD transgenic lines. In addition, we monitored phenotypic severity in four independent amiR zf-HD transgenic lines. To determine the correlation between the degree of molecular perturbation and phenotypic severity, we performed TaqMan qRT-PCR to measure amiRNA expression in leaf and floral tissues from each line. As shown in Figure 2d,e and Appendix A, higher amiRNA expression in floral and leaf tissues was associated with a stronger phenotype. Expression of target genes, including *HB23*, *HB24*, *HB26*, *HB27, HB34*, and *MIF2*, was lower in all four individual amiR zf-HD lines than in WT (Figure 2f and Appendix A). Expression levels of amiRNAs and their targets were correlated with the degree of phenotypic abnormality (Figure 2e,f and Appendix A). These results indicate that stronger molecular perturbation of ZF-HD TFs increases the severity of plant architecture phenotypes.

### 2.3. ZF-HD TFs Act as Central Regulators to Control Expression of Multiple TF Families

To elucidate the downstream targets and molecular pathways regulated by these ZF-HD TFs, we performed transcriptomic analysis in WT, amiR zf-HD, and *hbq* using Illumina RNA-seq. In all the *zf-hd* mutants, NF-YB, YABBY, Double B-box zinc finger (DBB), MYB-related, MIKC, and SBP TFs were downregulated, whereas GATA, GROWTH-REGULATING FACTOR (GRF), and B3 TFs were upregulated (Figure 3a–f). In particular, 23 out of 42 members of the MIKC gene family (also known as the MADS-box family), which are mainly involved in controlling flower and fruit development [37,38], were remarkably differentially expressed genes (DEGs) in *zf-hd* mutant backgrounds (Figure 3b and Appendix A). Among the B3 TF superfamily, reproductive meristem (REM) family genes, which are targets of floral identity genes [39], were also upregulated (Figure 3e). Taken together, these results indicated that ZF-HD TFs are central regulators that directly or indirectly regulate many TF families.

### 2.4. ZF-HD Gene Members Positively Regulate SPL Genes in a Feed-Forward Loop (FFL) via miR157

To elucidate the miRNA-mediated gene regulatory network, we performed small RNA (smRNA) profiling in young floral tissues to identify differentially regulated miRNAs between WT and *zf-hd* TF mutants. The miRNA expression patterns could be classified into several categories (Figure 4a,c,e–g). As expected, changes in miRNA expression in the mutants were reflected in changes in the levels of the corresponding target mRNAs (Figure 4b,d,e–g and Appendix A). In the *zf-hd* mutants, nine members of the *SBP* gene family targeted by miR156/7 (*SPL2*, *SPL3*, *SPL5*, *SPL6*, *SPL9*, *SPL10*, *SPL11*, *SPL13A*, and *SPL15*) were generally downregulated, whereas miR157 was consistently upregulated, consistent with the qRT-PCR data (Figure 4e and Appendix A). By contrast, based on the smRNA data, miR156 expression was unchanged in all mutant backgrounds. In contrast to miR157 activity, miR172 activity was downregulated in the amiR zf-HD and *hbq* with the expected upregulation of the miR172 target *TOE2* (Figure 4e,f and Appendix A). We did not exclude the possibility that the downregulation of miR172 was concomitant with the upregulation of the miR172 target *TOE2* and decreased expression of *SPL* which is upstream of miR172.

To systematically characterize miRNA-related biological network motifs in vivo, we integrated expression data from RNA-seq and smRNA data with transcriptional (ZF-HD TF and miRNA) and post-transcriptional (miRNA and miRNA target) [40] regulation data into a ZF-HD TF–related miRNA gene regulatory network, and used the integrated dataset to identify network motifs [41]. In comparison with a randomized network, the resultant network was enriched in type IV coherent FFLs (*p*-value = 0.0001 for amiR zf-HD, and 0.0002 for *hbq*) (Table 1 and Appendix A). Coherent FFLs have dynamic behaviors that function as a sign-sensitive delay [41,42]. We propose that the ZF-HD TF–related miRNA gene regulatory network, including the FFL comprising HB34, miR157, and SPL10, contributes to the regulation of plant architecture in *Arabidopsis*.

### 2.5. Genome-Wide Analysis of HB34 Binding

Given that HB34 was strongly expressed in young floral tissues (Appendix A) and downregulated in the four *zf-hd* mutants (Figure 2a–c), we hypothesized that HB34 is a key regulator of floral development. To identify the direct targets of HB34 at the genomic scale, we performed genome-wide chromatin immunoprecipitation sequencing (ChIP-seq) analysis of *HB34* in young floral tissue of *35S*:*HB34*-*GFP*-8-1 using an anti-GFP antibody (Appendix A). High-confidence binding sites of *HB34* were strongly enriched between the transcription start site (TSS) and 0.5 kb upstream, as well as between the transcription termination site (TTS) and 0.5 kb downstream (Figure 5a). Most (62%) of the binding sites were present in upstream regions (that is, within 3 kb of the TSS) (Figure 5b), and around half (1629 of 3217) of the candidate HB34 target genes had peaks located between +0.5 kb and −0.5 kb from the TSS (Figure 5a). To determine how these direct TF target genes relate to plant architecture, we combined the high-confidence ChIP-seq peak and DE TFs from RNA-seq data. The examination of expression patterns grouped by TF family revealed that DE patterns in certain TF families exhibited common trends (that is, up- or downregulation) in all *zf-hd* mutant backgrounds (Figure 5c,d and Appendix A). The three-amino-acid-loop-extension (TALE) and TCP TF families were enriched in modulated TF members: 6 out of 21 (28.6%) TALE TFs and 6 out of 24 (25%) TCP TFs were directly modulated. The TALE TF family mainly contains KNOTTED-like homeodomain (KNOX) and BEL1-like homeodomain (BELL) members [43]. In the modulated TALE TF family, five out of six were BELL members. TCP TFs were downregulated in amiR zf-HD and *hbq*. As shown in Figure 5e and Appendix A, 62–70% of direct targets of HB34 that were downregulated in mutants are associated with GO terms related to the regulation of cell death or phenylpropanoid biosynthetic process and upregulated genes related to plant organ development, indicating that HB34 positively regulates these processes. By contrast, HB34 generally acts as a negative regulator in plant organ development (Figure 5e and Appendix A). We performed computational prediction of putative HB34-binding motifs using position weight matrices (PWMs) and selected the top PWM (YTAATYAW) derived from the expressed dataset. The genome-wide distributions of HB34 binding sites relative to the TSS were derived based on these PWMs, and HB34 binding sites were enriched in the ~200 bp upstream of the TSS (Figure 5f,g and Appendix A).

To reveal the direct targets of HB34 in the miRNA regulatory network, we combined smRNA data, RNA-seq data for miRNA targets, and ChIP-seq data. Using this approach, we found that HB34 bound to the promoter of *MIR157D* and negatively regulated its expression. HB34 also bound to the promoter or 5′UTR of *SPL10*, a target of miR156/7, thereby positively regulating *SPL10* expression in young floral tissues in all the mutants. We conducted ChIP-qPCR to confirm the occupancy of HB34 in the *MIR157D* and *SPL10* promoter regions in the inflorescences of *35S*:*HB34*-*GFP*/*hb34*. We observed substantial enrichment of HB34 in the p2 promoter regions of *SPL10* and *MIR157D*, but not in the p1, p3, or p4 promoter region or p5 intragenic region, indicating that HB34 engages in direct and specific binding to these promoters (Figure 5m). Collectively, these relationships constitute a direct coherent type IV FFL mediated by miR157 that is involved in the regulation of plant architecture in *Arabidopsis*.

## 3. Discussion

The transcriptional and post-transcriptional gene regulatory networks that contribute to plant architecture remain largely undiscovered. In this study, we performed the first characterization of ZF-HD TFs using genetic and molecular approaches and defined the miRNA-mediated molecular mechanisms that determine plant architecture in *Arabidopsis*. Through a systems-biological approach using the amiR zf-HD and *hbq* mutants, we demonstrated that the network is enriched in feed-forward motifs. Accordingly, we propose the existence of a novel GRN in which HB34 directly represses the expression of *MIR157D* and cooperatively activates *SPL10*, thereby contributing to the determination of plant architecture.

Genetic analyses revealed that single loss-of-function mutants in ZF-HD TFs (*hb23*, *hb31*, *hb33*, and *hb34*) did not exhibit a phenotype, whereas multiple loss-of-function mutants (amiR zf-HD and *hbq*) exhibited disruptions in vegetative architecture (shoot branching and leaf size) and floral development. In addition, these analyses revealed that ZF-HD TFs play redundant functional roles during plant development. Furthermore, it remains possible that ZF-HD TFs interact with each other, thereby constituting a highly connected regulatory network [13,18]. The majority of MADS-box TFs are directly or indirectly regulated by ZF-HD TFs, and *AGL18* and *AGL20* are directly regulated by them (Figure 3 and Figure 5 and Appendix A). MADS-box TFs are involved in floral organ identity, flower development, flower timing, and determinacy of the meristem [45,46]. The observed alterations in expression might be responsible for the morphological phenotypes of *zf-hd* mutants, including abnormal flower shapes.

In *Arabidopsis*, regulation of shoot branching is controlled by a class II TCP transcription factor, *BRANCHED* (*BRC1*), an ortholog of maize *TEOSINTE BRANCHED1* (*TB1*), which promotes the accumulation of abscisic acid (ABA) [4]. Our ChiP and RNA-seq data indicate that HB34 directly binds to the promoter region of *BRC1* and activates its expression in young floral tissue (Appendix A). In addition, HB34 and other ZF-HD TFs activate HB-2, HB-5, and HB53 as downstream targets of BRC1 and directly regulate seven members of the HD-ZIP TFs (Appendix A). The NAC TFs and bZIP TFs ABSCISIC ACID RESPONSIVE ELEMENTS-BINDING FACTOR 3 (ABF3), G-BOX BINDING FACTOR3 (GBF3), ABA INSENSITIVE 5 (ABI5), and NAC-LIKE, ACTIVATED BY AP3/PI (NAP) of the ABA-related GRN were generally downregulated in amiR zf-HD and *hbq* (Appendix A). ABF3 is a master regulator of the ABA signaling pathway, and GBF3 and ABI5 are involved in the response to ABA [47,48]. NAP regulates ABSCISIC ALDEHYDE OXIDASE3 (*AAO3*), and the NAP–AAAO3 regulatory module controls ABA biosynthesis [49]. Therefore, ZF-HD TFs may play roles in the GRN related to ABA signaling. Additional high-resolution spatial and temporal transcriptome data combining ChiP-seq data will provide a detailed understanding of the relationship between HB34 and BRC1-related GRN with plant hormone and the response to light conditions.

In *Arabidopsis*, miR156/7 and miR172 play roles in the juvenile and adult stages of vegetative development, with miR172 acting downstream of miR156. Dynamic antagonistic expression of the miR156 and miR172 families controls the transition from the vegetative to reproductive stages [21,50]. Both the [44] miRNA156/7 and miR172 families are highly conserved across plant species [30]. Notably, in this regard, our smRNA profiling revealed elevated levels of miR157 and reduced levels of miR172 in the amiR zf-HD and *hbq* (Figure 4). These observations support the idea that the antagonistic expression of miR157 and miR172 contributes to plant architecture, and that HB34 represses miR157 while activating miR172.

Genome-wide binding analyses revealed that HB34 binds to the *MIR157D* promoter to negatively regulate the expression of miR157d, generating a feed-forward network motif with *SPL10*, a target of miR157. The detailed motif structure is a coherent type 4 feed-forward loop (C4-FFL); in such structures, a transcription factor X both directly activates the expression of target Z and represses Y, a repressor of Z [41,42]. Both HB34 (X) and miR157 (Y) are required for SPL10 (Z) expression. As we propose in our model (Figure 6), HB34 directly binds to *MIR157D* and *SPL10* and modulates their expression, thereby generating an FFL, and is also indirectly involved in similar FFLs that regulate *TOE2* via *MIR172A–C*. In addition, network motif analysis showed that coherent type IV feed forward motifs are overrepresented in the GRN involving HB34 relative to a random network (Table 1). FFLs are characterized by sign-sensitive delays and function in mammalian cell-cycle progression. In plants, FFL is involved in the cell death regulated by miR164 in *Arabidopsis* [41,51]. In the present context, we speculate that plant endogenous signals such as developmental stimuli can amplify *SPL10* expression by directly binding to the *MIR157D* promoter to inhibit the expression of miRNA157d in young floral tissues after the activation of HB34. This process involves a sign-sensitive delay. *SPL10* expression would not be activated until the level of miR157d reaches the activation threshold of the target gene, resulting in delayed action and persistent HB34 activation, not by short term activation, induced *SPL10* expression, indicating that the sign-sensitive delay could protect against short input fluctuation and achieve fine tuning and noise buffering of target genes.

HB34 does not directly bind to *MIR157A–C*, but nonetheless negatively regulates the expression of the corresponding miRNAs. It is possible that other ZF-HD TFs, or possibly TFs of other families, act as upstream regulators by binding to the *MIR157A–C* or *MIR156* promoter. Recently, AGL15 and AGL18 were identified as positive regulators of miR156; AGL15 binds directly to the CArG motif in the *MIR156A/C* promoter [52]. On the contrary, our genome-wide direct binding analysis identified the HB34-binding motif T(A/G)ATTA(A/G), which contains the tetranucleotide ATTA (TAAT on the complementary strand), which in turn resembles the core motif of the homeodomain (HD)-containing protein binding site [53]. In addition, we identified the YTAATYAW motif and found that its position frequency distribution graph closely matched the graph based on two other PWMs for HB34 (ZFHD_tnt.ATHB34_col_a_m1; ZFHD_tnt.ATHB34_colamp_a_m1) obtained from the publicly available Plant Cistrome Database (http://neomorph.salk.edu/PlantCistromeDB, accessed on 5 July 2020) [54], further validating the presence of this motif in the promoter region (Appendix A). The ZF-HD TFs HB23, HB24, HB25, HB33, and HB34 have similar core motif patterns containing the ATTA motif, although the core motifs of two other family members (HB21, HB32) are divergent [13,54].

These findings suggest that HB23, HB33, and HB34 share common direct target sites and may interact with other family members to regulate the expression of miR157.

The elucidation of the comprehensive GRN of a TF family enables prediction of the target genes of the members. In particular, we could use the miRNA mediated *HB34* module in crop species to identify candidate genes that could be used to modify branching and inflorescence architecture, supporting improved yield and biomass, thereby helping to address global food security and the growing demand for renewable energy resources. In addition, our expression analyses revealed that the downstream targets of these TFs include factors involved in cell wall formation, which is pertinent to biomass production. Therefore, we anticipate that ZF-HD TFs will be suitable for biotechnological applications, and the TF sub-network we describe here will provide insight into gene regulatory networks in other crop species.

## 4. Materials and Methods

### 4.1. Plant Growth and Mutant Collection

The *Arabidopsis thaliana* L. accession Columbia (Col-0) was used as the WT control for all mutant analyses. Plants were grown in a greenhouse at 22 °C under a 16 h light/8 h dark cycle. For root analysis, plants were grown vertically on 1× Murashige and Skoog salt mixture [55], 1% sucrose, and 2.3 mM 2-(N-morpholino) ethanesulfonic acid (pH 5.8) in 1% agar. Three T-DNA mutants, *hb23* (SALK_059288), *hb33* (SALK_097388), and *hb34* (SALK_085482C), were acquired from the *Arabidopsis* Biological Resource Center (ABRC), and the *hb31* mutant was obtained from the Sainsbury Laboratory *Arabidopsis thaliana* (SLAT) collection. For genotyping, genomic DNA was isolated using a modified CTAB (hexadecyltrimethylammonium bromide) method. Gene-specific primers were designed using the iSect Primers tool from T-DNA Express: *Arabidopsis* Gene Mapping Tool (http://signal.salk.edu/cgi-bin/tdnaexpress, accessed on 5 June 2014), and primers for the *hb31* lines were designed manually. We used WMD3 tools (http://wmd3.weigelworld.org/, accessed on 5 June 2014) to design amiRNA zf-HD lines to target nine ZF-HD TFs [33]: AT1G69600 (HB29), AT1G75240 (HB33), AT3G28920 (HB34), AT4G24660 (HB22), AT5G15210 (HB30), AT5G39760 (HB23), AT5G42780 (HB27), AT5G60480 (HB26), and AT5G65410 (HB25). Target sequences were TGGTGGAAGTTACGGTGGCAG (mir) and CTACCACCGTAACATCCACCT (mir*).

### 4.2. Localization of ZF-HD TF and Tobacco Transient Assay

Each TF gene (without a stop codon) was cloned into vector pMDC83 harboring the 35S promoter using LR Clonase. TF coding regions were C-terminally fused to GFP, and the fusion proteins were expressed under the control of the CaMV 35S promoter using the Gateway-Compatible Plant Transformation vector [56]. The resultant plasmids were transformed into *Agrobacterium tumefaciens* strain GV3101. *Agrobacterium* cells were grown at 28 °C for 2 days in YEP medium (5 g NaCl, 10 g Bacto Peptone, 10 g yeast extract) supplemented with rifampicin (60 mg/mL), kanamycin (50 mg/mL), and gentamycin (50 mg/mL). For transient expression analysis in tobacco, *Agrobacterium* were precipitated and re-suspended in infiltration buffer (10 mM 2-[N-morpholino] ethanesulfonic acid (MES) (Merck, Rahway, NJ, USA), pH 5.5, 10 mM MgSO_4_), and then used to inoculate the abaxial epidermal side of 3–4-week-old *N. benthamiana* leaves. Three days after infiltration, GFP signals in leaves were observed on an LSM710 confocal microscope (Carl Zeiss, Oberkochen, Baden-Württemberg, Germany).

### 4.3. Gene Expression and Sequence Analysis

For real-time qPCR, total RNA from the leaves and young floral tissues of *Arabidopsis* was extracted using TRIzol reagent (Ambion, Austin, TX, USA). RNA samples were treated with DNase I using the TURBO DNA-free kit (Ambion, Austin, TX, USA). cDNA was synthesized from 2 µg of total RNA using the Superscript III First Strand Synthesis kit (Thermo Fisher, Waltham, MA, USA). Each primer was designed to amplify a 200–300 bp region of the corresponding transcript, and the *UBI10* transcript was used as the normalization control for qPCR. All real-time PCR analyses were performed on a C1000 Thermal Cycler (Bio-Rad, Hercules, CA, USA). Primers were tested on WT Col-0 by quantitative PCR using the Bio-Rad CFX96 Real-Time System to check primer quality, as determined by melting curve analysis and amplification efficiency. Gene expression was measured in three technical replicates for each sample; for these experiments, cDNA was generated by reverse transcription of 2 µg of total RNA, as described above. Primer sequences used for qPCR are provided in Appendix A. Protein sequences were obtained from Gramene BioMart using protein domain IPR006456 with manual curation, and homology was evaluated using Ensembl Compara GeneTrees [57].

### 4.4. miRNA Expression Analysis by TaqMan PCR

To determine the effects of miRNAs, we used the stem-loop reverse-transcription protocol (TaqMan assay) to compare mature miRNA expression levels between wild type and *zf-hd* mutants. Primers were designed using UPL RT primer [58]. snoR41 was used as a normalization control. For each expression experiment, TaqMan assays were performed in two technical replicates for the wild type and each mutant. All real-time PCR analyses were performed on a C1000 Thermal Cycler.

### 4.5. RNA-seq Data and smRNA Library Data Analysis

Using the TRIzol method, total RNA was isolated from WT and mutant flower tissues and roots from plants grown in a greenhouse or growth chamber. RNA quality was checked on an Agilent Technologies 2100 Bioanalyzer, and RNA-seq libraries were prepared using the Illumina TruSeq RNA Sample Prep kit (RS-122-2001, Illumina, San Diego, CA, USA). Small RNA libraries were prepared following the TruSeq Small RNA Library Prep Kit (RS-200-0012, Illumina). Libraries were generated by Macrogen, and sequencing was performed on the HiSeq 2000 platform (Illumina, San Diego, CA, USA). Expression analysis of roots and flower tissues was evaluated using Tophat for mapping and Cuffdiff for differential expression analysis, both with default parameters [59,60]. The TAIR10 genome sequence (www.arabidopsis.org, accessed on 5 June 2014) was used as the reference genome, and the TAIR10 genome annotation was used as the gene model with no novel junctions. For differential expression analysis on WT flower and mutant flower tissues, DESeq was used with default parameters [61]. For small RNA analysis, reads were adapter-trimmed and filtered for length and quality, and then mapped onto the TAIR10 genome sequence using Bowtie with options “-v 0 -m 40 -a --best --strata” [62]. Read count tables for each miRNA (5p/3p) and its precursors were generated using htseq-count (http://www-huber.embl.de/users/anders/HTSeq/doc/count.html), using the chromosomal coordinates of *Arabidopsis thaliana* microRNAs from miRBase (http://www.mirbase.org) v19 as the model and option “-m intersection-nonempty”. Library size was normalized against the total number of 18–30-nt small RNA reads in each library. Differential expression of mature miRNAs and precursors were analyzed using DESeq with default parameters [59]. GO term analysis was performed using agriGO with singular enrichment analysis (SEA), and the Bonferroni option was used for multiple testing correction [63].

### 4.6. ChIP-seq Analysis

For ChIP-seq, we constructed transgenic plants expressing HB34 fused to GFP expressed under the control of the 35S CaMV promoter, *35S*:*HB34*-*GFP*, and confirmed the expression of *HB34*-*GFP* (Appendix A). A homozygous transgenic line was selected for subsequent use. Chromatin immunoprecipitation (ChIP) was performed as described by Bowler et al. with the following modifications [64]. ChIP-seq libraries were constructed for use with the NEXTflex CHIP-seq kit (Illumina-compatible). Reads were mapped to the TAIR10 reference genome using Bowtie (Langmead et al., 2009), and the MACS v1.4 peak caller [65] was used with default parameters to identify peaks for which HB34 binding was significantly higher than the background level. Overlapping peaks with summits within 600 bp and non-overlapping peaks with summits within 300 bp were merged. High-confidence peaks were assigned when nearby peaks were observed in two biological replicates and filtered to those in promoter regions of annotated genes. A total of 3217 genes with high-confidence peaks (*p*-value < 1 × 10^5^) from 3 kb upstream of a transcription start site (TSS) or 1 kb downstream of a transcriptional termination site (TTS) were identified as candidate HB34 targets.

### 4.7. Motif Prediction Analysis

We derived the top PWM from the experimental data (ChIP-seq and RNA-seq) using MEME-ChiP [66]. In addition, we obtained the two PWMs of the ATHB34 binding site from the publicly available Plant Cistrome Database (http://neomorph.salk.edu/PlantCistromeDB) [54]. We used these three PWMs to computationally predict the over-represented transcription factor binding sites (TFBS) in the promoter regions 1000 bp upstream and downstream of the TSSs of expressed genes using Search Tool for Occurrences of Regulatory Motifs (STORM) from the Comprehensive Regulatory Element Analysis and Detection (CREAD) suite [67]. To determine the background cut-off, the same analysis was performed on a random set of genomic sequences. Motifs were considered overrepresented if they appeared more often in promoter sequences of protein-coding genes than in a background set of the same number of random genomic sequences (*p*-value < 0.001).

### 4.8. Network Motif Analysis

We identified four types of FFLs involving the ZF-HD TF HB34 by combining our expression data with miRNA targets obtained from a literature review and psRNATarget [68]. In our three-node FFLs, the first node represents HB34, the middle node represents a DE miRNA, and the third node represents a DE miRNA target. Assuming that the miRNA always has a repressive effect on the target, there are four types of FFLs in which the miRNA can be fixed in this position: Incoherent type 1 (I1), Incoherent type 2 (I2), Coherent type 3 (C3), and Coherent type 4 (C4). To calculate *p*-values for the FFL types, we shuffled the expression values of the miRNAs and target genes 10,000 times and compared the randomized FFL distributions.

## 5. Conclusions

Our data points to a novel feed-forward loop between a ZF-HD, miR157, and its target genes, in which loss of function leads to increased branching and inflorescence architecture. In this study, we identified a new function of ZF-HD34 (HB34): direct regulations of miRNA157D and its target *SPL10*. We found that ZF-HD TFs, especially HB34, negatively regulate the expression of miR157 and positively regulate *SPL10*. Genome-wide ChIP-Seq analysis revealed that *miR157D* and *SPL10* are direct targets of HB34, creating a feed-forward loop that constitutes a robust miRNA regulatory module. Network motif analysis contained overrepresented coherent type IV feedforward motifs in the amiR zf-HD and *hbq* mutant background. This finding indicates that miRNA mediated ZF-HD feedforward modules modify branching and inflorescence architecture in *Arabidopsis*. Taken together, these findings not only reveal a guiding role of ZF-HD TFs in the regulatory network module but also demonstrate its role in plant architecture in *Arabidopsis*. Furthermore, the model herein provides insights into candidate genes that determine agronomic traits associated with yield.

## Figures and Tables

**Figure 1 ijms-23-08665-f001:**
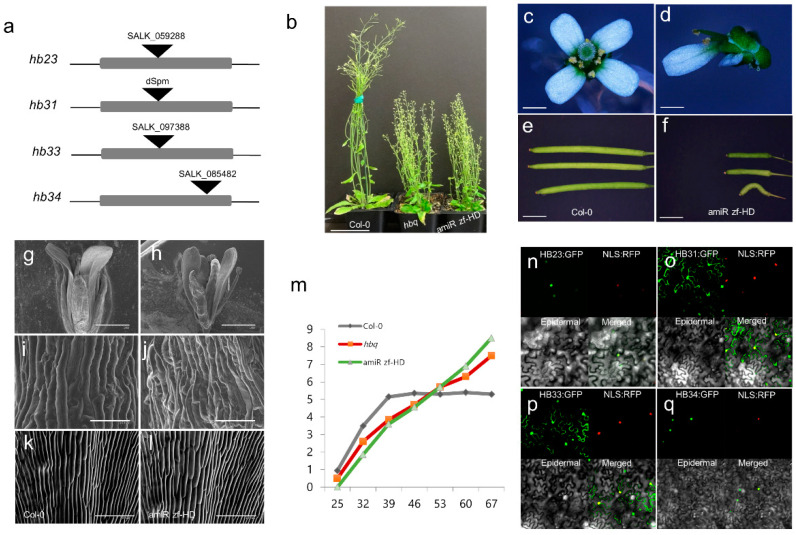
amiR zf-HD and *hbq* lines exhibit plant architecture phenotypes. (**a**) T-DNA insertion sites in *HB23*, *HB33,* and *HB34* and dSpm transposon insertion in *HB31*. (**b**) From left to right: mature Col-0 (Wild Type, WT), *hbq,* and amiR zf-HD; scale bar: 5 cm. (**c**–**f**) Floral architecture in Col-0 (**c**) and amiR zf-HD (**d**); bar: 1 mm. Silique shape in Col-0 (**e**) and amiR zf-HD (**f**); bar: 1 mm. (**g**–**l**) Scanning electron microscope picture showing floral morphology in Col-0 and amiR zf-HD. Flower in Col-0 (**g**) and amiR zf-HD (**h**); scale bar: 1 mm. Sepal in Col-0 (**i**) and amiR zf-HD (**j**); bar: 100 µm. Proximal part of petal in Col-0 (**k**) and amiR zf-HD (**l**); bar: 100 µm. (**m**): Lateral branching number of Col-0, *hbq*, and amiR zf-HD, during development. X axis: days after germination. Y axis: number of branches. (**n**–**q**) Confocal images of localization of HB23 (**n**), HB31 (**o**), HB33 (**p**), and HB34 (**q**) in tobacco epidermal cells in the transient expression assay. GFP is green, and NLS-RFP is red.

**Figure 2 ijms-23-08665-f002:**
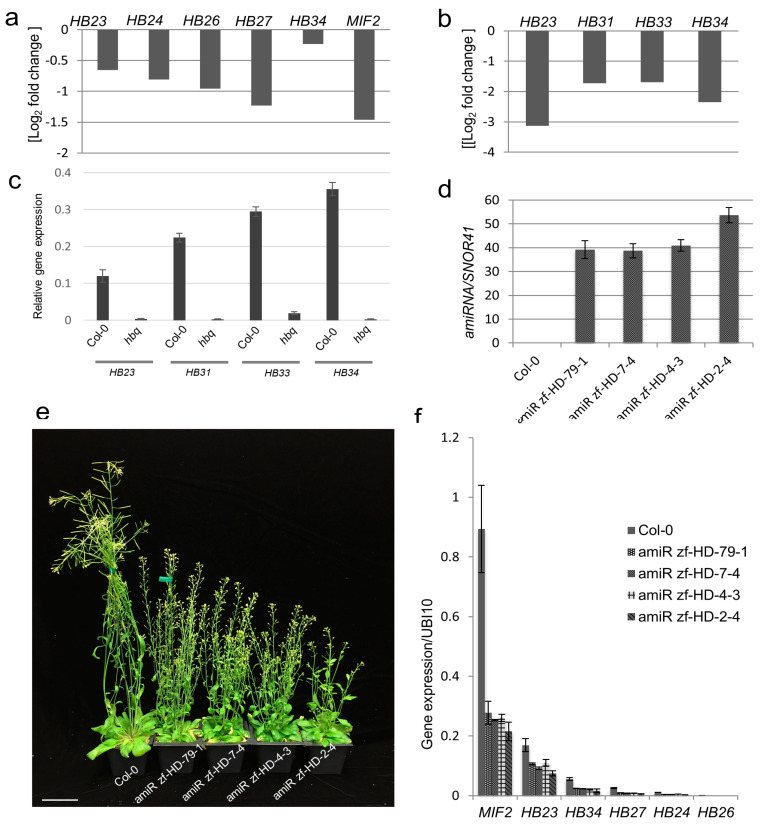
Phenotypes of *zf*-*hd* mutants are the result of disturbed expression of ZF-HD TF genes. Expression of ZF-HD TFs in amiR zf-HD (**a**) and *hbq* (**b**), as determined by RNA-seq, in comparison with WT (*p* value < 0.05). (**a**,**b**) Three biological replicates were used for RNA-seq and figures shown as average of expression value (Log2 fold change). *p* value refers to the cutoff for determining whether the genes were significantly differentially expressed in the mutant relative to WT. It is determined by the differential expression analysis software based on the expression levels observed across replicates. (**c**) Expression of *HB23*, *HB31*, *HB33*, and *HB34* was monitored by qRT-PCR in Col-0 and *hbq*. (**d**) Expression of amiRNA in young floral tissue in four independent homozygous transgenic lines: amiR zf-HD−79−1, amiR zf-HD−7−4, amiR zf-HD−4−3, and amiR zf-HD−2−4. (**e**) Phenotype of mature Col-0 plant and four individual amiR zf-HD transgenic lines. Scale bar: 5 cm. (**f**) Target expression of amiRNA in four individual amiR zf-HD transgenic homozygous lines (amiR zf-HD−79−1, amiR zf-HD−7−4, amiR zf-HD−4−3, and amiR zf-HD−2−4) relative to WT.

**Figure 3 ijms-23-08665-f003:**
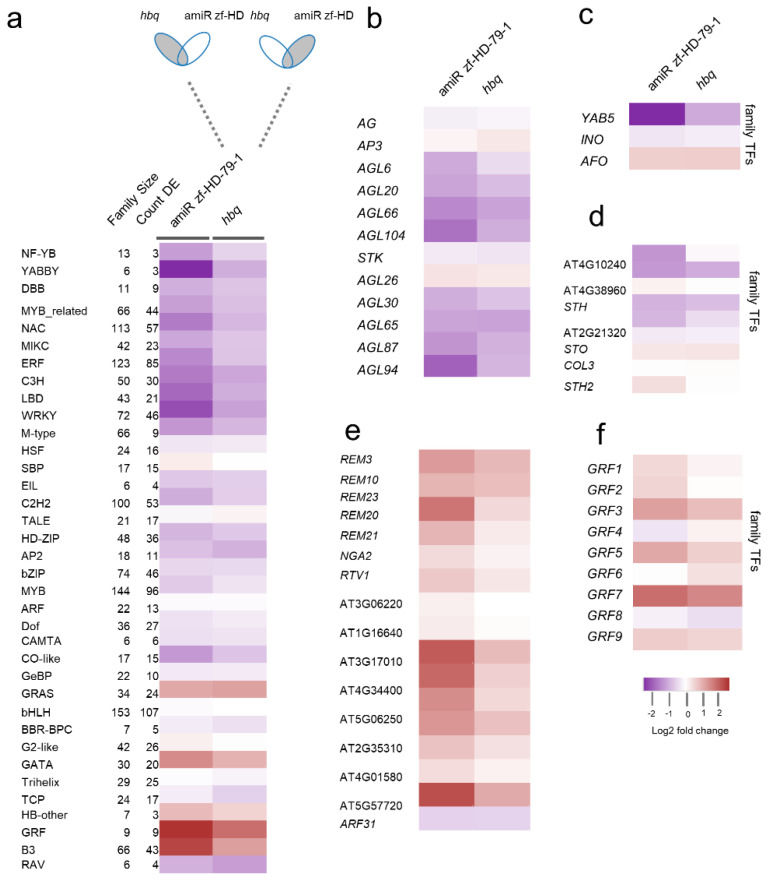
Heat map of TFs differentially expressed in young floral tissue in comparison with WT, as classified by TF family. (**a**) Expression profile of genes, grouped by TF families. The Venn diagrams above the columns are shaded gray to indicate the co−occurrence of differentially expressed genes in response to amiR zf-HD and to *hbq* mutants. Three biological replicates were used for RNA-seq. From left to right: genes DE in amiR zf-HD and *hbq*; in amiR zf-HD and *hbq*; in *hbq*; and in amiR zf-HD. (**b**–**f**) Individual TF family genes with altered expression patterns in *zf*-*hd* mutants: (**b**) MIKC and M−type (MADS−box) TFs, (**c**) individual (YABBY) TFs, (**d**) double B−box zinc finger (DBB) TFs, (**e**) B3 TFs, and (**f**) GROWTH−REGULATING FACTOR (GRF) TFs. Three biological replicates were used for RNA-seq.

**Figure 4 ijms-23-08665-f004:**
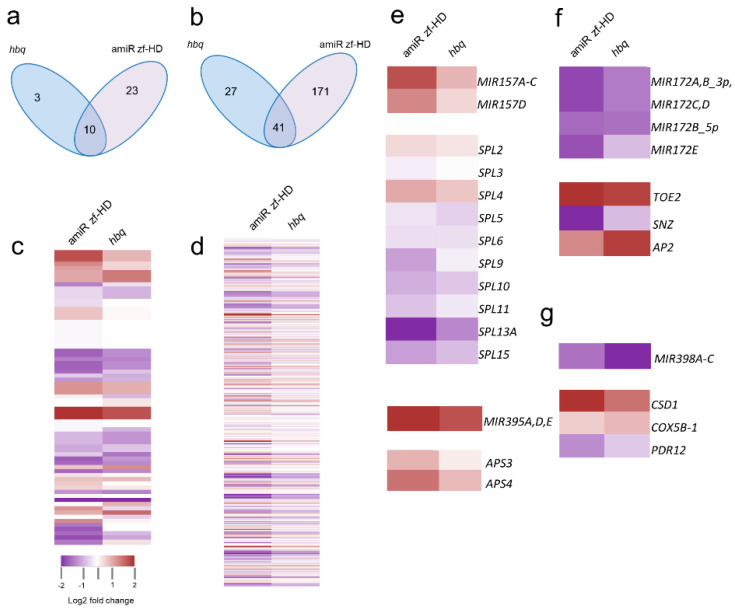
The ZF-HD–mediated miRNA gene regulatory network is enriched in FFLs. (**a**–**d**) smRNA profiling of miRNAs and miRNA target perturbation in *zf-hd* mutants in comparison with WT. (**a**,**b**) Venn diagram of DE downregulated miRNAs (**a**) and their targets (**b**). (**c**) Heat map of smRNA analysis in young flowers of amiR zf-HD and *hbq,* relative to WT. (**d**) Heat map of miRNA targets in young flowers of amiR zf-HD, and *hbq*, in comparison with WT. (**e**) Heat map of category I, which includes miR157 and miR395 and their targets (bottom). (**f**) Heat map of category II, which includes miR172 and their targets. (**g**) Heat map of category III, which includes mir398 and their targets.

**Figure 5 ijms-23-08665-f005:**
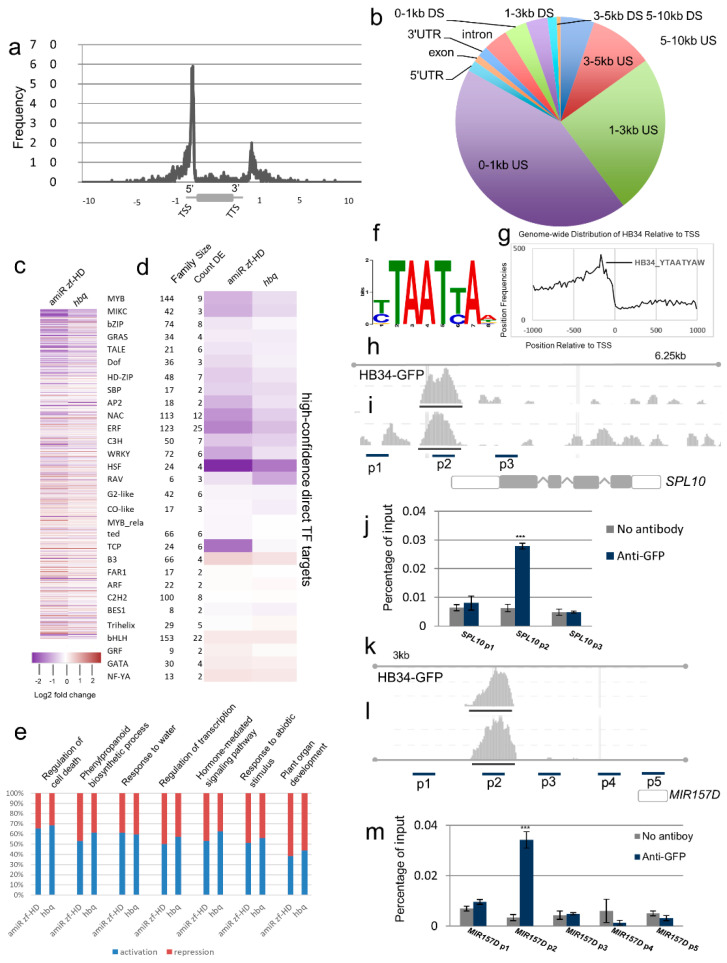
Genome−wide binding target analysis of HB34 reveals that HB34 directly regulates *MIR157D* and *SPL10* via an FFL. (**a**) Frequency of HB34 binding sites in *Arabidopsis* gene models revealed strong enrichment in the regions within ±0.5 kb of transcription start and stop sites. (**b**) Genome-wide distribution of binding peaks in genic regions, DS: downstream, US: upstream. (**c**) Heat map of direct HB34 targets with altered expression in *zf*-*hd* mutants. (**d**) Heat map of average gene expression of direct HB34 targets, grouped by TF family. (**e**) Relative proportion of activated/repressed genes associated with enriched GO terms among the directly bound targets of HB34. (**f**) Genome browser view showing HB34 ChIP−seq reads aligned near *SPL10* and regions p1, p2, and p3 selected for ChIP−qPCR validation. (**g**) ChIP−qPCR analysis of HB34 in the *SPL10* promoter region and p1, p2, and p3 regions as depicted in panel f. (**h**,**i**) Genome browser view showing HB34 ChIP-seq reads aligned near *MIR157D* and regions p1−p5 regions selected for ChIP−qPCR validation. (**j**) ChIP−qPCR analysis of HB34 in the *MIR157D* promoter region. (**k**,**l**) Analysis of HB34 binding sites from ChIP-Seq and RNA-seq datasets revealed enrichment of a novel motif (YTAATYAW). (**m**) Genome−wide distribution of the novel motif YTAATYAW at specific positions relative to the TSS of genes with HB34 binding sites. (**j**,**m**) The result is mean ± SD (Student’s *t*-test, *p* < 0.001 ***).

**Figure 6 ijms-23-08665-f006:**
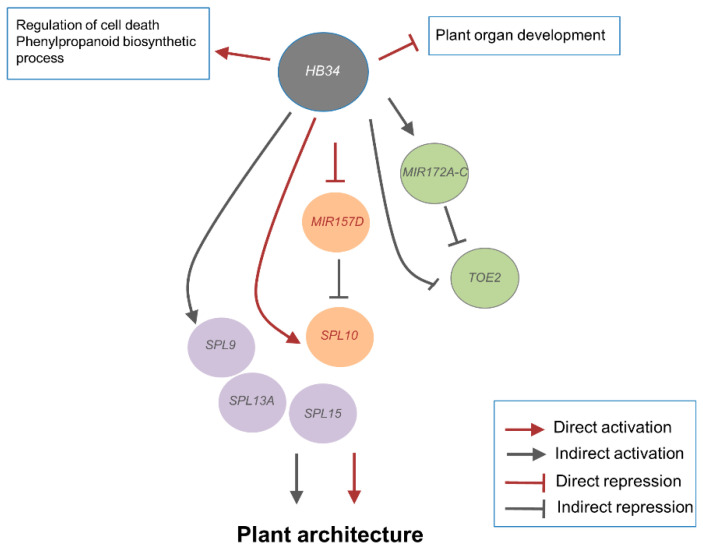
Proposed model of regulation mediated by HB34 and miR157. HB34 binds and represses *MIR157D* to activate *SPL10*, generating an FFL, and indirectly regulates the *SBP* gene family (*SPL9, SPL13A*, and *SPL15*). HB34 also indirectly activates *MIR172A–C*, which are involved in the FFL with *TOE2*, a target of miRNA172. Based on the expression profile and mutant phenotype analysis, HB34 is involved in various developmental processes: it is a negative regulator of plant organ development and a positive regulator of cell death and phenylpropanoid biosynthesis.

**Table 1 ijms-23-08665-t001:** Network motif analysis comparing Col-0 and *zf-hd* mutants.

miRNA–Target Feed-Forward Motif	Number of Loops in Real Network (Average Number of Loops in Randomized Network)		Number of Loops in Real Network in *p*-Value	
mutant	amiR zf-HD	*hbq*	amiR zf-HD	*hbq*
Incoherent type I	20 (13.54)	3 (4.0769)	NS (0.1182)	NS (0.7195)
Incoherent type II	21 (15.9115)	0 (2.3681)	NS (0.1930)	NS (1.0000)
Coherent type III	20 (15.8256)	5 (3.8105)	NS (0.2298)	NS (0.3346)
Coherent type IV	41 (13.6795)	14 (2.5448)	*** (0.0001)	*** (0.0002)
Total	102 (58.9525)		22 (12.8003)	

Significant differences are represented by asterisks; *** *p*-value < 0.001, Number of randomizations in randomized network is 10,000. The statistical test used was a randomized permutation test [44] to determine how often a randomized network has more motifs of each type than the experimentally determined network. The statistical test performed to calculate a *p* value was a randomization test involving *n* = 10,000 randomized networks. The *p* value is the frequency of a random network having more motifs of a given type than the input network.

## Data Availability

The data presented in this study are available in insert article or Appendix A here.

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
