# Peer review of "Role of a ZF-HD Transcription Factor in miR157-Mediated Feed-Forward Regulatory Module That Determines Plant Architecture in Arabidopsis"

_ijms, 2022, doi:10.3390/ijms23158665_

Round 1

Reviewer 1 Report

The manuscript is valuable and interesting. It is well written and organized. The topic fits within the scope of the journal. I only have a few minor suggestions that should be considered before publishing the text as listed below.

-         Please refer to full botanical names of species when mentioned for the first time, e.g., Oryza sativa L.

-         The Abstract is missing a conclusion.

-         Keywords should be arranged alphabetically and do not repeat words from the title.

-         The name of the genus should be written in italics, e.g. line 46.

-         The English form is generally good. Only minor punctuation errors require correction, e.g., line 52.

-         Murashige and Skoog salts lack a reference

-         Lines 451-452: city, country?

-         Please follow the MDPI formatting style.

-         Please complete information on the producers of key chemicals and equipment used, including the name of the producer, city, state, and country.

-         Line 506: reference style.

-         I suggest updating some of the older references. Currently, only four of the 66 papers cited were published in the past five years.

-         Some of the photos lack scale bars.

-         Lines 217-218 - ?

-         Line 329: according to what statistical test? What was the “n” number?

-         A separate conclusions chapter should be provided.

-         The abbreviation list at the end of the manuscript is missing.

After incorporating all the necessary changes, the manuscript can be accepted for publication.

Author Response

We thank the reviewer for the recommendation. All of the recommended changes have been incorporated in the revised manuscript.

Reviewer 2 Report

The authors describe genetic and genomic analyses of ZF-HD gene family function in arabidopsis. Using loss of function mutants and amiRNA approaches, they convincingly show that ZF-HD genes act redundantly to regulate plant architecture, in particular branching and floral development. Plants are stunted and more branched, and floral development is perturbed, in the zf-hd lines. Using genome-wide transcriptome sequences the authors identify a number of gene families that are misregulated in zf-hd lines, members of some of which have previously been shown to regulate branch and flower development. To investigate potential regulatory mechanisms, the authors take a ChiP-seq approach and show that the ZF-HD gene HB34 directly binds multiple targets across the genome. Integrating their results, the authors conclude by describing a speculative model of how HB34 regulates plant architecture via-MIR157D.

Although the plant phenotypes are interesting and the genome-wide data is of merit (although how important the RNA-seq of floral tissue is to the role of ZF-HDs in shoot branching is unclear), the authors’ interpretation of the data is questionable. In particular, their focus on miR157 as a critical determinant of HB34 function. They demonstrate that the primary transcripts of miR157 genes are increased in zf-hd mutants, and that HB34 binds to a region upstream of the MIR157D sequence. However, previous analyses of miR157 activity (i.e. He at al 2018 Plos Gen) have revealed that miR157 has a relatively small role in plant architecture. They also conclude that the increase in miR157 in the zf-hd is reflected by the decrease in expression of SPL genes, which are targeted by miR157. However, SPL genes are largely translationally regulated by miR157. SPL3, which shows the greatest response to transcriptional cleavage, is almost unaffected in the RNA-seq survey. Furthermore, is unclear how functionally significant the single HB34-binding region is in the MIR157D gene. Likewise MIR157A-C, which are generally thought of as more functionally significant than MIR157D, and were more significantly upregulated in the zf-hd lines, do not appear bound by HB34. A more convincing interpretation of the results is that HB34 regulates multiple SPL genes independent of miR157. Further genetic (i.e generation of hbq; mir157d/spl10 mutants) and biochemical (i.e. removal of the HB34 binding domain from the MIR157D sequence) analyses would be highly useful in testing the authors model. Without it, the extensive speculation on feed-forward loops and genetic pathways is unmerited.

The rationale and experimental detail are often vague and the paper would benefit from further editing. Specific comments are below. In addition, the quality of some of the images need to be improved prior to publication, particularly figures 1d, S2b. For a paper on floral phenotypes it would be good to have photos of full flowers.

Line 81: The description of CUC function is repeated and is not helpful to the reader and could be simplified

Line 84-90: The introduction would benefit from a brief description of the role of the miR156/SPL module in branching control.

Line 91: Clarify that HB34 is a ZF-HD protein. The results that follow also suggest a redundant role for multiple ZF-HD proteins, rather than specifically HB34.

Line 101: Why the authors chose to focus on miRNAs is unclear

Line 105: How closely related are these 4 ZF-HD genes? The manuscript would benefit from a tree of the gene family. Conclusions about the roles of the gene will vary considerably depending on the relationships within the gene family.

Line 110-112: ‘slightly’ is vague. According to the Fig S1 the authors found statistically significant differences.

Line 116: “hbq plants had a more pronounced morphological phenotype than the hb33 hb34 double mutant” – this statement needs a figure reference

Line 118: ‘multiple ZF-HDs were targeted’ – be specific about the targeted genes

Line 135: What happened to HB23?

Line 146: “we observed 146 strong GFP fluorescence in the nucleus” – the expression of HB23-GFP doesn’t look very strong to me

Line 169: To demonstrate specificity of the amiRNA approach, if the authors have genome-wide data it would be useful to show the expression for all ZF-HD genes rather than a selection.

Line 194: Do any of the lines correspond to the amiRNA ZF-HD data described previously? If the previous amiRNA analyses use a specific transgenic line it should be named.

Line 243: miR172 regulates TOE2 via translation repression and there you would not expect to see any effect of miR172 at the transcriptional level. The downregulation of miR172 is instead ‘concomitant’ with the decrease in SPL gene expression, which are upstream activators of miR172 expression.

Line 285: Does this include TCP genes that are known to regulate branching?

Figure 1: the text refers to 1b-f, 1k-p, 1q and then 1g-j. Figure 1 needs to be re-arranged so it reflects the order of the text.

Figure 1 legend: “amiR zf-HD, hbq, and the two repressor lines exhibit plant architecture phenotypes” – what are the two repressor lines?

Figure 2: c – swap HB23 and 31 to be consistent with other graphs. A. p-value is referred to in the legend but no instances of statistical significance are shown in the graphs. Was this overlooked? The type of statistical tests should also be referred to in the legend.

Figure 3: a red-green colour scheme will be difficult to interpret for colour-blind readers. I have no idea what the Venn diagrams above a) represent.

Figure 4: c,d should be accompanied by names of gene families. E) would benefit from being split between MIR157 and MIR395. E-g contain multiple false ‘targets’ of the miRNAs. This is highly misleading. Only direct targets of the miRNAs should be included.

Figure 5: the two browser tracks in h and j need labelling

Figure S3: It is unclear from both the Figure and the Supp methods how many replicates were carried out for the RNA-seq analysis. If more than 1 the variance should be indicated on the graph, if only 1 these data need to be removed.

Author Response

We thank the reviewer for the consideration. All of the recommended changes have been incorporated in the revised manuscript.
